# Using Hybrid Telepractice for Supporting Parents of Children with ASD during the COVID-19 Lockdown: A Feasibility Study in Iran

**DOI:** 10.3390/brainsci10110892

**Published:** 2020-11-22

**Authors:** Sayyed Ali Samadi, Shahnaz Bakhshalizadeh-Moradi, Fatemeh Khandani, Mehdi Foladgar, Maryam Poursaid-Mohammad, Roy McConkey

**Affiliations:** 1Institute of Nursing Research, University of Ulster, Newtownabbey BT37 0QB, UK; r.mcconkey@ulster.ac.uk; 2Raha Autism Education and Rehabilitation Center, Tabriz 51368, Iran; moradi.1151@yahoo.com; 3Fariha Autism Education and Rehabilitation Center, Tehran 25529, Iran; drkhandani@autismfariha.ir; 4Ordibehesht Autism Education and Rehabilitation Center, Isfahan 83714, Iran; Mehdi.fouladgar@yahoo.com; 5Daily Rehabilitation Centre Section, Iranian State Welfare Organization (ISWO), Tehran 25529, Iran; marypoury77@gmail.com

**Keywords:** telepractice, autism spectrum disorders, COVID-19, parental-mediated intervention, Coronavirus, daycare center

## Abstract

During the three-month closure of clinics and day centers in Iran due to the coronavirus disease 2019 (COVID-19) lockdown, parents of children with Autism Spectrum Disorder (ASD) became solely responsible for their care and education. Although centers maintained telephone contact, it quickly became evident that parents needed more detailed advice and guidance. Staff from 30 daycare centers volunteered to take part in a two-month online support and training course for 336 caregivers of children with ASD of different ages. In addition to the provision of visual and written information, synchronous video sessions were used to coach parents on the learning goals devised for the children. Both qualitative and quantitative data were collected to understand the acceptability of using telepractice and the outcomes achieved. A low dropout rate and positive feedback from parents indicated that they perceived telepractice sessions to be useful. The factors contributing to parents’ satisfaction were identified. Although the use of telepractice would be a good alternative for caregivers in any future lockdowns, it could also be used in conjunction with daycare center services to encourage greater parental participation, or with families living in areas with no day centers. Further studies are needed to compare telepractice to usual daycare face-to-face interventions, and to document its impact and cost-effectiveness for parents and children.

## 1. Introduction

Communication technology used by healthcare professionals is diverse [1]. Modern technologies offer a range of flexible modalities, ranging from simple daily applications (e.g., phone calls, email, and voice and video messaging) to complex technologies (e.g., interactive web-based software and interactive virtual classrooms). Smartphones, tablets, and laptops are more generally accessible to the general population at a reduced cost [2]. A recent review [3] explored the increasing usage of technology as a viable option in providing home health education and counseling to various populations in need of support. The uptake of these technologies has been lower in-person education, therapy, and social services—especially in services for children with special needs, who are mostly dependent on face-to-face interactions. Nevertheless, Camden et al. [4] concluded in their review that “available communication technology might be particularly well suited to implementing best practices for children with disabilities when the focus of the therapies is on supporting the children and their families, problem-solving with them to foster the child’s development and functioning” [4].

Despite the dearth of applications in technologies for children with developmental disabilities, the available studies report promising results for children with Autism Spectrum Disorder (ASD) [5,6]. ASD is a neurodevelopmental disability that affects social communication and behavior development, and it typically manifests in the early stages of life [7]. Improved parental knowledge, parental intervention fidelity, and improved social behavior and communication skills for children with ASD were reported in a review of 15 studies on parent-mediated intervention training delivered remotely [8,9]. Moreover, parents appreciated being active agents in this approach; it gave them access to appropriate training and ongoing guidance so that they were able to deliver the intervention in a consistent manner [10]. Furthermore, evidence in using communication technology for individuals with ASD (at different age levels) is emerging, with preliminary findings suggesting that it has potential benefits in service delivery and cost savings, such as speedier set-up and coverage in rural areas [11,12].

### 1.1. Telepractice

The term telepractice is a general term that embraces other terms, such as telehealth and telemedicine, and has been defined as “the application of telecommunications technology to deliver professional services at a distance by linking service provider to a client, or supervisor to service providers for assessment, intervention, and/or consultation” [13]. Two approaches for delivering telepractice are defined as synchronous and asynchronous. When the telepractitioner and client are in a one-to-one or group setting, and are interacting in real-time via video and/or audio, this is referred to as a synchronous telepractice. Asynchronous telepractice occurs when information, such as videos, pictures, or audio files are recorded and exchanged via technology between the telepractitioner and client (and vice versa), with no live interaction between them. This approach is known as “store and forward”. When both synchronous and asynchronous methods are used in combination, this is referred to as hybrid telepractice, which combines the benefits of both synchronous and asynchronous approaches.

### 1.2. Supporting Families in Iran

In Iran, there has been very limited use of telepractice with families of children with ASD. Rather, the focus has been on the preparation of written and visual materials, such as those produced by the first (SAS) and the last (RM) authors and their colleagues, which arose from a series of research studies into the needs of Iranian parents [14,15,16]. The tailored, parent-focused program that the mentioned authors devised [17,18] was based on the biopsychosocial model of disability. There were five booklets on different aspects of ASD, in lay language, plus a toolkit consisting of eight practical booklets to enhance parental reciprocity and interactive communication through everyday life activities and play. The booklets offer simple information and practical advice to enhance communication and its development in different stages. Simple, self-completed checklists were provided to parents, so they could have a better understanding of their child’s level of functioning. Modifications to the home environment, to address the child’s sensory preferences, along with strategies to manage unusual behaviors, were also covered in the booklets. It was also envisaged that the booklets would enable and empower parents to nurture their child’s development at home, alongside the teaching and therapy the child would receive at school or a daycare center. Fortuitously, this resource was available when the COVID-19 lockdown commenced.

The Iranian Social Welfare Organization (ISWO) provides at least 110 daycare centers, across the 31 provinces in Iran, for children with autism spectrum disorder (ASD). The centers care for children aged 3 to 14 who usually attend, daily, under the supervision of these centers. The centers provide educational and rehabilitation services, and are open from 08.00 a.m. to 12.00 p.m. (4 h), to provide a wide range of daily services, which are mainly sponsored by the government. Most of the centers also provide afternoon extracurricular and rehabilitation services funded by parental fees. As the world became increasingly affected by the COVID-19 pandemic, Iran followed the advice of WHO [19] and UNESCO [20] in closing all of the educational and daycare centers (the closures started in March and lasted until May 2020). Telecommunication through mobile-based technology was the only possible approach to deliver professional services at a distance, by linking daycare centers to caregivers for assessment, intervention, and/or consultation.

### 1.3. Country Profile

The prevalence of ASD in Iranian children was reported to be 6.26 per 10,000, which is lower than that reported for some Western nations, but in line with rates from other countries [21]. ASD services in Iran fall under the Ministry of Health and Ministry of Social Welfare, with 90 percent of healthcare services provided through governmental services [22]. Improvements in recent decades have resulted in healthcare services covering the majority of the population [23]. The Iranian Social Welfare Organization (ISWO) provides clinical and daycare services to preschoolers with physical and intellectual disabilities, and older children with developmental disabilities, who were assessed by educational services as not suitable to attend mainstream schools. The government pays the attendance expenses for the majority of families, while other parents pay for a portion of the services they receive, based on their socioeconomic situation. Moreover, children with developmental disabilities who attend mainstream schools may also come to ISWO centers after school to receive therapy and specialist interventions from psychologists and therapists. Families contribute to these costs. The maximum capacity of daycare centers, based on the approved regulations in Iran, is fifty children. Iran ranks first in terms of the number of people in the Middle East with access to telecommunication services and satisfactory internet infrastructure, with an estimated 43 million users [24]. The widespread use of smartphones, social media [25], and computers throughout Iran enabled some daycare centers for adults (with mostly physical disabilities) to provide video-conferencing, to deliver rehabilitation services alongside consultations, advice, and guidance to their service-users [26].

### 1.4. Developing a Hybrid Telepractice for Families

The ISWO recognized the need for daycare centers for children with ASD to stay active, and to continue providing support for family caregivers who were in desperately need of assistance due to their around the clock caregiving because of the COVID lockdown. The only possible way was to use technology and available telecommunication services. Most daycare centers had already established mobile phone-based groups using Telegram or WhatsApp channels, in which they provided caregivers with one-way, non-interactive, and passive forms of daily center information, and news sharing. However, neither the daycare centers nor caregivers were ready for the newly imposed roles placed on them, but there was no other choice available to reduce the danger to children and families negatively impacted during the lockdown. ISWO quickly took the decision to pilot the use of telepractice services from their day centers by using mobile phone-based video technology. Caregivers during the lockdown would be observed in their homes while interacting with their children, with remote supervision and coaching provided by staff from the daycare center.

Previous studies on developing telepractice services have stressed the crucial preparatory role of organizational processes, in providing support and resources to prepare therapists and practitioners in implementing new models of practice [27]. However, the COVID-19 imposed closure of centers shortcut these preparations; thus, telepractice services had to be developed and implemented in a very short period using existing resources, such as the parent training resources as described above. More positively, family caregivers, who had previously been reluctant in engaging with their child’s education and therapy, were asking the (then closed) daycare centers for practical advice and guidance for managing their child at home. This confirmed the assertion by Chorpita et al. [28] that parental advocacy and training is the most important element in the successful implementation of new models. ISWO invited the first author (SAS) to oversee the development of the telepractice materials and to supervise their staff in implementing them. SAS had previously acted as the senior consultant with the ISWO on ASD research and training courses and was familiar with the staff and service users. The brief was to test the feasibility of using mobile phone-based telepractice services in a parent-implemented, home-based intervention program, under the supervision of the daycare center staff.

The feasibility study addressed three main questions.

Is telepractice a feasible approach for providing services to family caregivers and children with ASD in a less affluent country such as Iran?What are the factors that contribute to caregivers’ positive attitudes regarding the telepractice services provided to them and their children with ASD, in the absence of in-person daycare center services?Is it possible to increase the effectiveness of telepractice services for caregivers of children with ASD?

The present study was carried out over eight consecutive weeks—telepractice services in 30 daycare centers across the country with a maximum capacity of 50 individuals with ASD admission. This report can be considered a proof-of-concept study, in that it examines how telepractice was developed and implemented in a home setting through continuous support from a daycare center. It also provides a foundation in which further studies can be built regarding the effectiveness of telepractice.

## 2. Methods

### 2.1. Setting up the Telepractice Service

Three parties were engaged in the telepractice program: i) the family caregivers; ii) day center staff, and a course supervisor with extensive experience of ASD; and iii) ISWO centers in Iran who provided support and supervision to the day centers. Following a literature review of existing telepractice studies involving families of children with ASD, the heads of 50 day centers (in Iran) for children with ASD were invited by the course supervisor to join an online discussion group, in which they were invited to share their perceptions on parental information and support needs during the closure of centers. The discussion, using online synchronous and asynchronous focus groups, continued for over one week, and resulted in a listing of priority issues required to implement a telepractice service, which included the following:Suitable resource materials from Iran—written and visual—were identified to act as a guide for center staff, as well as for sharing with caregivers as appropriate.Each participating center nominated a key person as the main coordinator of the center’s telepractice. In most instances, this was a person with the required qualification to supervise the daycare center’s daily services. During the lockdown, other center staff were involved with caregivers and children on a scheduled daily basis for routine contact, but the key person’s responsibility was the coordination, supervision, and monitoring of the prepared online telepractice program.An online group was created for the course supervisor and daycare center staff for them to develop procedures relating to freeing up time from other clinical work; making different reading materials accessible for parents, the provision of high-quality supervision and training, establishing peer-learning working groups and planning periodic evaluation of the program.Identifying and creating video-based parental training materials for use alongside written materials. Videos are reported to be more effective [29].Caregivers needed to have smartphones or similar devices with home internet access and the freeware program, WhatsApp version 4.0.0 (Mountain View, California, 2009), with the free calling feature. This app was also used for documents and link sharing, online video calls, observing the home session, and coaching the parent.

The daycare centers provided online support in two forms, group and individual sessions, which were scheduled based on the center and parental preferences. Both individual and group sessions used the hybrid approach of support to provide both synchronous and asynchronous sessions.

#### 2.1.1. The Main Aims of the Telepractice Service

The telepractice model for parent-implemented, home-based interventions was based around an online, daily, hybrid telepractice training session, for caregivers or parents, administered by the daycare center’s key person. The aims and objectives were:To devise individual learning plans for a child with ASD in conjunction with caregivers to use at home.To boost the confidence of caregivers in managing their child with ASD at home.To answer caregivers’ questions through the provision of accurate personalized information.To provide updated information relating to ASD.

The link below covered different areas of caregiving, with the main focus on communication and behavioral management in the natural home setting, through play, with a focus on daily living. For each part, there were separate tutorial videos, along with written and oral information, and self-rated scales, which was shared with caregivers. (http://www.behzisti.ir/news/12221). Parents were guided in the use of structured teaching, behavioral approaches, and environment modifications, which were adapted to the child’s communication and sensory preferences, based on a parent-implemented intervention perspective.

#### 2.1.2. Implementing the Telepractice Service

During the training sessions, daycare center staff aimed to encourage parent–child interaction in a modified natural home environment, using behavioral and structural strategies based around common pictures and objects. Training sessions for each child were developed and documented in a weekly training plan, focusing on communication, and sensory and cognitive domains. Caregivers were encouraged through video clips, pictures, and printed sources to replicate the program at home, and to make video recordings of interactions with their child. The daycare center’s key person provided parents with feedback, cues, and coaching for the proper implementation of the intervention strategies. All of the home-based sessions were monitored by the center’s key person, who in turn submitted fortnightly reports to the course supervisor (SAS). SAS also provided regular coaching and training to the key persons in each center.

In addition, there was also a virtual meeting place for key persons across the centers for social networking, exchanging information, and for contacting the therapists and clinicians to answer questions. All of these sessions were digitally video-recorded by each daycare center for later analysis.

#### 2.1.3. Evaluating Telepractice

To allow for a more thorough understanding of the impact of telepractice on caregivers, a mixed-methods approach was used [30]. To date, concerning research on parent-mediated intervention, telepractice has generally adopted a quantitative approach, whereas parental and practitioner perceptions, as the main stakeholders of this service’s delivery model, have yet to be examined as a primary outcome variable [31]. Therefore, caregivers who finished the course, as well as those who dropped out, were asked for their feedback regarding the course, and its shortcomings, via a WhatsApp questionnaire consisting of closed- and open-ended questions, which were possible for them to answer using voice messages or in written form. Caregivers who completed the online course were invited to answer six open-ended questions: 1.What is the most important advantage of the online training course?2.What is the most obvious shortcoming of the online training course?3.If you have to continue using online courses for a long time, what are your recommendations for improvement of the quality of the course?4.Which part of the information was most useful for you?5.Which part was less useful for you?6.Do you have any further comments about the course?

Caregivers who dropped out were also asked about the reason for their leaving. In all, 15 (36%) voice recorded messages were transcribed along with 27 (64%) written comments, verbatim. A thematic content analysis approach was used to analyze the responses [32].

In addition, quantitative data were collected using pre- and post-course design. Assessment measures administered at pre- and post-course were as follows: a researcher-made questionnaire, regarding a main parental complaint about caregiving, online course, information provision, and the level of provided support. The video analyses were assessed based on a fidelity rubric, which considered the following 10 items—the most common and neglected items that might happen in a training or communicational session with children with ASD. The items were: (1) consistency with the environment (considered for a different type of training), (2) environment modification and stimulus control, (3) providing visual notification about the possible uncontrolled stimulus, (4) not forcing the child to do requests, (5) following the child’s comfortable position, (6) understanding the child’s reaction, (7) using visual icon for the start, (8) creativity in using toys and play, (9) parental temper control, using the visual icon to notify, and (10) finishing the task. Each item was rated on a five-point scale. After the course, caregivers were allowed to rate the course and the daycare center, to rate the parental level of engagement using a Likert scale, and to evaluate their experience as providers of intervention services to their child.

All procedures in the present study were in accordance with the ethical standards of the ISWO All caregivers signed an online consent form in which their rights to confidentiality, and to withdraw from the project at any stage of the study, were mentioned.

### 2.2. Participants

Parents and daycare center key persons were the key stakeholders in this feasibility study: namely 30 daycare coordinators from ASD centers and 336 caregivers of children with a confirmed diagnosis of ASD through the professionals of ISWO.

#### Key Persons

The 30 daycare centers were located in 19 (61%) of the 31 provinces across the country. The demographic information on the 30 key workers is presented in Table 1.

Concerning caregivers and children, Table 2 gives the demographic details of 417 (28%) out of 1500 caregivers (the maximum number of parents based on the registration volume permission granted by ISWO) who initially volunteered to be enrolled for the telepractice (an average of 11 caregivers participated from each center), and contrasts the 336 caregivers who completed two months of the online training course of the center, in which their child was registered, and the 81 caregivers (19%) who registered, but failed to complete the telepractice sessions. In Table 3, demographic information of children who both completed the online course and who dropped-out are presented.

### 2.3. Activity Records

The number of days daycare centers offered individual services to family caregivers differed across centers (Mean = 5.60, days SD = 1.47 Max = 7, Min = 2) as did the number of daily hours the centers spent for each family and child (Mean = 1.20 h SD = 0.40 Max = 2 Min = 1). The number of days in which daycare centers were active each week over the eight weeks differed across centers (Mean = 5.36 days SD = 1.67 Max = 7 Min = 1). The number of hours spent on each call also varied (Mean = 1.70 h SD = 0.79 Max = 4 Min = 1). During the eight-week presentation of the course, the course supervisor was available for support in a WhatsApp group, in which all 30 daycare centers were members. An average of six contacts per centers in each week was recorded

## 3. Results

### 3.1. Qualitative Findings

At the end of the study period, all caregivers were invited to send their feedback on the effectiveness of the telepractice. Responses were received from 42 caregivers (12.5%) who finished the telepractice and 8 (9.9%) from caregivers who dropped out. In all, 30 responses came as written comments and 20 as voice messages.

All caregivers thought that the new mobile-based social media facilities were user-friendly and easy to use. A mother said (No.1): “Getting access to the course contents and contact with others through my mobile was not a challenge at all. This is turned to be a part of our daily life”. 

Caregivers generally said that the most important advantage of the course, for them, was to give caregivers a hand, when they were in the utmost need, with continuous caregiving. A father said: (No. 39) “Being engaged with my son and helping him to progress by my energy and at my pace while being directly engaged in the process and having more hope that I wanted for him in such a sudden unpredictable hard time. Thanks”.

Almost all caregivers found no parts of the provided information useless and some mentioned special rehabilitation or educational information as being most applicable. They recommended different issues to improve the services, but mainly suggested hard copies and more video resources in the form of training packages. A mother said, “Internet package in the form of low price internet from Telecom system, and more hardcopy info before the course in CD and DVD format will be very helpful” (No. 3). Similar suggestions and recommendations were repeated in response to the final question. A mother said: “We have always been engaged with our children but not in a systematized way and with considering aims and objectives and under the guidance of professionals. Keep this good job continue. This was excellent” (No. 43).

Of the 41 caregivers who completed the telepractice, 38 (93%) caregivers thought that they would continue with the service, and would consider it as one of their choices, as well as recommend it to other caregivers as a very useful service. However, three caregivers (7%) were reluctant to continue the telepractice service because of the extra financial demands that it imposed on them, or technical problems, such as the internet speed. A father from the completed training course group said: “We had a serious problem with the internet and extra expenses we had to pay to top up the system on a weekly basis. This is important for us to be cautious about the extra expenses in this economically difficult time. I am unemployed because of the COVID-19 now” (No. 27).

The reasons given by caregivers who dropped out included the following. Some were not persuaded that online services were sufficient for children with ASD and their caregivers. Other issues were raised, such as extra pressure being placed on caregivers, or it being a beneficial service only for the daycare center, as they were still entitled to receive governmental financial assistance, while the pressure was mainly on caregivers instead of daycare provider centers. A mother from the dropout group said: “I do not approve of this online system. You are getting governmental financial help to work with our children not to force us to do it by ourselves” (No. 49).

Some undesired aspects of online services were mentioned, including the sharing of videos and pictures of the children, even if was assured that they would not be used or seen by the others.

A mother from the dropout group said: “Open the daycare centers. I do not want to take videos of my child and to share it online to be seen by the entire world!” (No.19).

Parents who dropped out were asked for their suggestions regarding the training courses for them or their children. They mostly requested for the reopening of the daycare centers, or requested private home services. A mother said, “Just try to make safe places at school and reopen the centers as before” (No. 50). A father from the same group said: “I think you should look at your services to cover a wide range of children with Autism. What was offered was not suitable for all. These children are unique. He was also critical of the amount of information caregivers were asked to provide. I think you wanted to test a new service for us. I am a scholar myself and familiar with these activities. You forgot about the service and paid more attention to the data you wanted to collect. You cannot test a service while you are providing it” (No. 30).

### 3.2. Quantitative Findings

Daycare centers asked caregivers about the main difficulties regarding caregiving during the lockdown. They also rated parental perception of the severity of ASD in caregivers through a self-rated scale. Key persons also rated the parental level of engagement, in the process, through their level of activity and their provision of requested records. Parental satisfaction with the support course, after the course, was evaluated, as was caregiver and daycare center staff attitudes to the online course.

Parental reaction to telepractice: caregivers were asked to rate their perception of online training courses by choosing between three choices: positive, negative, and having no ideas. These ratings were repeated at the end of the telepractice session. Before the course started, 7.4% of respondents rated it as positive; after the course, this had risen to 61.0%—a statistically significant change in attitude (chi-square (4) = 71.16, *p* < 0.001).

Parental reactions were further investigated in relation to the children’s characteristics. Parents of the younger children were more satisfied with the course (86%) than those with older children (28%) (chi-square (2) = 1.17, *p* < 0.001. Moreover, caregivers whose child with ASD had another accompanying diagnosis were less positive about the course (52%) than those with a single diagnosis of ASD (68%) (chi-square (2) = 9.79, *p* = 0.007).

At the outset, younger aged parents were more positive (10%) than were older parents (2%) about the telepractice course (chi-square (2) = 21.15, *p* < 0.000), but afterwards, the percentages of positive ratings had reduced in younger parents (68%) and increased for older parents (51%) (chi-square 2) = 10.68, *p* = 0.005).

Likewise, caregivers who had assistance with caregiving were more positive regarding the online course at the outset (11%) compared to those without assistance (0.5%), but for both sets of caregivers, these percentages rose to 71% and 47%, respectively, although they were still statistically significant (chi-square (2) = 23.57, *p* < 0.001).

Comparing children’s gender indicated that girls were more likely than boys to have dual diagnosis (48% of boys compared to 62.5% of girls (chi-square (1) = 6.51, *p* = 0.007). There was no statistical significance reported between the child’s gender and the birth order.

Parents concerns: caregivers’ specific concerns were grouped into behavioral, communication/talking, restlessness (including difficulty keeping their child inside), and a combination of all the areas. After the lockdown, and before the telepractice course started, the percentage of parents reported each type of concern was: Behavior (51%), Communication (14%), Restlessness (9%), and All areas (61%). After the course, the percentages had changed significantly: Behavior (12%), Communication (35%), Restlessness (17%), All areas (16%). Table 3 summarizes these findings. Concerns about behavior had reduced markedly (t = 10.67, *df* = 335 *p* < 0.001); while “has concerns in all areas” was (t = 18.35: *df* = 335 *p* < 0.001); whereas concerns about communication (t = 6.43, *df* = 335 *p* < 0.001, had increased and, to a lesser extent, so had restlessness (t = 2.65, *df* = 46, *p* < 0.05).

Caregiver dropout: considering 8 years as the cut off for the child age, it showed that parents of older children (33%) were more likely to leave the course than parents of young children (6%) (chi-square = 48.19, *df* = 1 *p* < 0.000). Similarly, the drop rate was higher for parents whose child had an additional diagnosis (29%) compared to those with a single diagnosis (9%) (chi-square = 26.08, *df* = 1, *p* < 0.000).

Fidelity checks: both the key person and caregiver fidelity in implementing the suggested practices were monitored using two specially developed rubrics. Parental fidelity scores were rated by the key person on each center on a four-scale rating, from weak to excellent. In all, 177 (42.4%) out of 336 caregivers were rated as excellent, and only 2 (.5%) were rated as weak. The key person’s fidelity on the same rating scale rated by the course supervisor showed no center staff was rated as weak, with 47% (14 daycare centers) rated as excellent. Caregivers’ with higher fidelity ratings (81%) had more positive attitudes to the online course than those with lower fidelity scores (33%) (chi-square (6) = 74.18”*df* = 6, *p* < 0.000).

A significant relationship was also seen between levels of the key persons’ fidelity score and previous participation in the ASD professional training course presented by ISWO (64% vs. 53% who did not participate and were cored as excellent based on the fidelity form), chi-square (2) = 8.32, *p* = 0.016. Although it was expected that key persons who scored higher in the fidelity rubric were more likely to have caregivers with higher scores of fidelity, the correlations between these fidelity ratings were not strong, although it was nearly statistically significant (Spearman correlation Rho = 0.33 *p* < 054.).

## 4. Discussion

This is one of the first studies to investigate the use of telepractice with families of children with ASD in a low resource country. Results indicating that a hybrid model of telepractice supervised by staff from daycare centers might be considered as useful support for children with ASD and their families in times of continuous caregiving due to situations, such as the COVID-19 pandemic. Caregivers can be guided to become effective teachers in the child’s most natural environment of the home. Such an approach can enable ISWO to better fulfil its remit of supporting children with ASD and their families.

Regarding the question of the feasibility of the telepractice, the findings indicate that telepractice could be a feasible approach for certain caregivers of children with ASD in particular [34,35]. Findings indicate that updated and trained daycare staff using remote access via a smartphone can enter the caregivers’ living places, and coach them while they are actively caregiving in their natural environments. Moreover, the telepractice services enabled the home environment to come under professional observation, at little expense, and without the time and effort involved in making visits to the family home.

Caregivers’ overall satisfaction and positive attitudes to the online course allied to a relatively low level of dropouts also indicate the feasibility of this service. This engagement has also increased parental knowledge about the main challenges in taking care of their children, such as managing their child’s behavior while also highlighting the difficulties around communication. This helped them to focus on increasing their nonverbal-communication skills in their interactions with the child [36].

However, there is a need for the daycare center staff to have training and support throughout the implementation of telepractice, as this approach requires them to have different skills, knowledge, and commitment. Those who participated in the previous ASD training course, provided by the course supervisor, were more successful in course implementation compared to those who did not. Hence, staff training and preparation should be considered as a key element in successful enactment of telepractice services.

Regarding the second question of this study which was searching for elements that contribute to caregivers’ positive attitudes regarding the telepractice services, several factors contributed to the caregivers’ satisfaction with online services. Younger caregivers were more optimistic about using telepractice; similar to that found in other services for children with ASD in Saudi Arabia: a similar culture to this sample [37]. Thus, telepractice might be targeted more at younger parents with younger children [38].

It was also found that having assistance at home is a good indicator of caregiver satisfaction with the online courses; presumably, because they had extra help at home and could devote more time to their child [31]. Consideration might then be given to the provision of more online support services directed at the carer’s needs, such as sharing their parenting stresses and experiences with other parents involved with the course.

The third question of this study which was searching for ways to improve telepractice and boosting effectiveness of this service, some of which have been noted already. However, it is likely that hybrid approaches that combine face-to-face contacts alongside telepractice would be better suited to some parents, especially those unfamiliar or reluctant to use technology. Moreover, the lack of internet access, or its associated costs, are also factors that limit the use of telepractice, especially for less affluent families and those in rural areas. Moreover, although telepractice may seem a possible solution for families in more remote areas who receive no support, issues around the availability of smartphones and internet access will need to be resolved first.

Nevertheless, caregivers should be reassured that telepractice is not considered a substitute to in-person daycare services, as this was echoed in the comments from some parents who dropped out of the course. Rather, it provides a means for ensuring that the training presented in the daycare centers can be extended into home settings of children with ASD. Moreover, parents who had higher fidelity scores in implementing the advice they were given were more supportive of telepractice, a finding that has been previously noted [39]. A follow-up study over an extended data collection period is needed to monitor the level of fidelity in the implementation of the strategies used by the caregivers at home, after their involvement in an online course. This would also help to determine ways of sustaining their engagement in home-based activities.

Cost-benefit analyses need to be undertaken in terms of financial costs and staff time and to compare the outcomes with the cost-benefits of face-to-face support by therapists and day centers. In addition, the development of multi-media, telepractice support courses on specific topics should be considered as an efficient means of sharing knowledge with family caregivers.

Finally, there were some limitations to the present study. It had to be prepared in an emergency and it took place over a limited period. More parents might have dropped-out if it went on for a longer period. It was not possible to recruit a control group, which had received similar services in a face-to-face situation because of the lockdown. Moreover, this was a self-selected group of parents and the findings need to be replicated with a more representative group of families whose children attend the day centers.

Although this feasibility study demonstrated that telepractice applications hold promise as a way of addressing some of the caregivers’ challenges during a time of a permanent caregiving situation, there is still lack of evidence for understanding the possible harms and limitations of the telepractice, and the way that various rehabilitation, assessments, and training protocols may be used through telepractice. Further studies needed to identify the caregivers and types of services in which a telepractice delivery system is appropriate or not. Such comparative studies will enable service providers to select a telepractice delivery model tailored to specific subgroups to maximize the benefit for them. Moreover, studies should focus on approaches to develop online support systems in developing countries, with the limitation of accessibility of services in remote areas, especially the rural parts in general [40].

It goes without saying that helping caregivers become capable members of the service intervention teams, involved with children with ASD, necessitates considerable specialized training in a wide range of domains for those leading the teams. The lack of highly trained professionals in different disciplines involved with children who have ASD is a major impediment in less affluent countries. Perhaps telepractice courses for clinicians developed internationally could help overcome this deficit.

## 5. Conclusions

With the increasing prevalence rate of ASD globally, service systems in less affluent countries face extra challenges in meeting the needs of caregivers and individuals with ASD. The telepractice model that has been tested with a sizeable number of families across Iran provides some basic evidence to support its potential to address some of the challenges associated with caregiving for children with ASD, even though it may not suit all parents. Telepractice, via telecommunication and mobile-based services, should be considered as a valuable adjunct to the current models of service provision in Iran and internationally. Further research is needed on the issue of COVID-19 and its impacts on children with ASD, their caregivers, support, and service, or possible alternative treatments not necessarily in the context of telepractice.

## Figures and Tables

**Table 1 brainsci-10-00892-t001:** The key persons’ demographic data.

Variable	
Gender	Male 5 (17%)
	Female 25 (83%)
Age	Mean (37.10) SD (6.32)
	(Min 25 Max 55,)
Education	Undergraduate 5 (17%)
	Graduate 22 (73%)
	Postgraduate 3 (10%)
Profession	Psychologist 19 (63%)
	Occupational Therapist 5 (18%)
	Speech and Language Therapist 2 (7%)
	Educational Science 3 (10%)
	General Health 1 (3%)
Experience with ASD in years	Mean (8.26) SD (3.23)
	(Min 1, Max 15)

Twenty-one (70%) of the key persons had already participated in the Iranian Social Welfare Organization (ISWO) professional training courses for Autism Spectrum Disorder (ASD) [33], although 9 (30%) had not.

**Table 2 brainsci-10-00892-t002:** Demographic data of caregivers who completed the online course and the dropout groups.

Variable	Completed Course Group	Drop Out Group
	N = 336	N = 81
Relationship with the child with ASD	Mother: 279 (83%)	Mother: 57 (70%)
	Father: 17 (5%)	Father: 12 (15%)
	Sibling: 9 (3%)	Sibling: 4 (5%)
	Grandparent: 1 (0.3%)	Grandparent: (−%)
	Both Parents: 30 (9%)	Both Parents: 8(10%)
Caregivers age	Mean (35.79) SD (6.51)	Mean (37.88) SD (6.87)
	(Max 70, Min 18)	(Max 56, Min 22)
Caregivers education in years	Under-university education: 210 (62.5%)	Under-university education: 57 (70%)
	University Education: 126 (37.7%)	University Education: 24 (30%)
Caregivers Profession	Housewife: 216 (64%)	Housewife: 54 (67%)
	Public work: 60 (18%)	Public work: 14 (17%)
	Technician: 26 (8%)	Technician: 6 (7%)
	Education: 16 (5%)	Education: 3 (4%)
	Medical and Health: 14 (4%)	Medical and Health: 4 (5%)
	Unemployed: 4 (1%)	Unemployed: (−%)
Having assistance with caregiving from the family members	Yes: 192 (57%)	Yes: 43 (53%)
	No: 144 (43%)	No: 38(47%)

**Table 3 brainsci-10-00892-t003:** Demographic data of children who completed the online course and the dropout groups.

Variable	Completed Course Group	Drop Out Group
	N = 336	N = 81
Children’s Age	Mean (8.06) SD (2.78)	Mean (10.81) SD (2.31)
	(Max 14, Min 3)	(Max 14, Min 3)
Children’s Gender	Boys 261 (78%), Girls 75 (22%)	Boys 60 (74%), Girls 21 (26%)
Birth Order	First born: 203 (60%)	First born: 47 (58%)
	Second born: 102 (30%)	Second born: 29 (38%)
	3rd and above born: 31 (10%)	3rd and above born: 5 (4%)
Children’s diagnosis	ASD: 158 (55.5%)	ASD: 19 (23.5%)
	Dual Diagnosis (diagnosis of ASD and other impairments such as Attention Deficit and Hyper Activity (ADHD), Cerebral Palsy (CP), or Intellectual Disability ID): 151 (45%)	Dual Diagnosis: 62 (76.5%)

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
