# Peer review of "Using Hybrid Telepractice for Supporting Parents of Children with ASD during the COVID-19 Lockdown: A Feasibility Study in Iran"

_brainsci, 2020, doi:10.3390/brainsci10110892_

Round 1

Reviewer 1 Report

I thank the editor for the possibility to review this article. 

The author proposed a protocol for telemedicine applied during the lockdown for COVID19. Families with a child with a ASD diagnoses were recruited. 

The theoretical background is not more deepened, but was satisfying. 

The cultural framework description was good. Although the protocol require further validation studies, I think it is more useful for clinical settings. 

I suggest several revisions before the publication.

A. Introduction section:

  • I believe it is necessary a brief description of the ASD condition referred to DSM-5 diagnostic criteria. 
  • The author did not report references about “available studies report promising results” (line 49); please, report them; furthermore, I suggested the following papers which applying the technologies to improve social competences of children with autism. It hope they will be useful for author.

Leo, M., Distante, C., Member IEEE, Mazzeo, P. L., Carcagnì, P., Levante, A., Petrocchi, S., Lecciso, F. (2019). Computational Analysis of Large-Scale Visual Data for Quantifying Facial Expression Production. Applied Sciences, 9, 4542. Doi: 10.3390/app9214542.  

Manzi, F., Peretti, G., Di Dio, C., Cangelosi, A., Itakura, S., Kanda, T., ... & Marchetti, A. (2020). A robot is not worth another: Exploring children’s mental state attribution to different humanoid robots. Frontiers in Psychology, 11, 2011.

Leo, M., Distante, C., Member IEEE, Spagnolo P., Mazzeo, P. L., Carcagnì, P., Rosato, A. C., Petrocchi, S., Pellegrino, C., Levante, A., Lecciso, F. (2018). Computational Assessment of Facial Expression Production in ASD Children. Sensors, 18, 1-25. Doi: 10.3390/s18113993.  

The introduction section reported the aim of the study. Authors referred to them as “questions”, in the discussion section they referred to the as “aim”. I suggest to standardize the label to use: research questions or aims?

I did found a ethical committee approval. Did the authors require an approval? Did parents signed a consent form? 

B. Method

The section is well organized. I think it would be interesting to report the intervention methods following by the children. Is it a psychomotor intervention? Is it ABA intervention? 

I suggest to rename the para 2.2.2 as ”Caregivers and children” since authors reported both parents’ and children’s socio-demo info. 

Regarding the children’s socio-demo info, it is not clear to me what “double diagnosis” means. What is the second diagnosis besides autism? 

C. Results

With regard to the qualitative findings, I did not understand if and what kind of statistical analysis were carried out. Why the authors did not run a content analysis? Or cluster analysis? Which criterion led the choice of the sentences reported in the text? 

Regarding the quantitative analysis, I suggest to summarized several results in a table. It will provide the results more readable. 

Furthermore, I believe that the way to report the chi-squared and t-test results were not correct. For example, Chi sq (df) = value; p = value. Also, when author reported the t-test results, they removed the df value. 

D. Discussion and conclusion section

In the conclusion section was missed the prevalence rate of the autism and references. 

In general, I suggest to revise the punctuation in all text. For example, there are several typos in the use of bracket, point, and semicolon. 

Author Response

We appreciate the useful comments we received from both reviewers and in the following table we have done all our best to meet them all.

Reviewer 1

A. Introduction section:

I believe it is necessary a brief description of the ASD condition referred to DSM-5 diagnostic criteria.

Following paragraph has been added:

Line 52-53 page 2:

ASD is a neurodevelopmental disability that affects social communication and behavior development and is typically manifests in the early stages of life[5]

The author did not report references about “available studies report promising results” (line 49); please, report them; furthermore, I suggested the following papers which applying the technologies to improve social competences of children with autism. It hope they will be useful for author.

The following references have been added to  line 49 page  

Vismara, L.A., McCormick, C.E., Wagner, A.L., Monlux, K., Nadhan, A. and Young, G.S., 2018. Telehealth parent training in the Early Start Denver Model: Results from a randomized controlled study. Focus on Autism and Other Developmental Disabilities, 33(2), pp.67-79.

Narzisi, A., 2020. Phase 2 and Later of COVID-19 Lockdown: Is it Possible to Perform Remote Diagnosis and Intervention for Autism Spectrum Disorder? An Online-Mediated Approach. Journal of Clinical Medicine, 9(6), p.1850.

The introduction section reported the aim of the study. Authors referred to them as “questions”, in the discussion section they referred to the as “aim”. I suggest to standardize the label to use: research questions or aims?

aim has been substituted for question  in the manuscript (page 11 line 417)

B. Method

I did found a ethical committee approval. Did the authors require an approval? Did parents signed a consent form?

Following paragraph was added to the bottom of the Method part:

All procedures in the present study were in accordance with the ethical standards of the ISWO.  All caregivers signed an online consent form in which their rights to confidentiality and to withdrawal at any stage of the study were mentioned. 

The section is well organized. I think it would be interesting to report the intervention methods following by the children. Is it a psychomotor intervention? Is it ABA intervention?

The following sentences was added to page 5 line 212:

Parents were guided in the use  of structured teaching, behavioral approaches and environment modifications which were adapted to the child’s communication and sensory  preferences.  based on a parent-implemented intervention perspective.

I suggest to rename the para 2.2.2 as ”Caregivers and children” since authors reported both parents’ and children’s socio-demo info.

The subtitle has been changed to

Caregivers and children

Regarding the children’s socio-demo info, it is not clear to me what “double diagnosis” means. What is the second diagnosis besides autism?

In the table the term dual diagnosis is explained as meaning  Autism along with other diagnosis such as ADHD, CP, or ID

C. Results

With regard to the qualitative findings, I did not understand if and what kind of statistical analysis were carried out. Why the authors did not run a content analysis? Or cluster analysis? Which criterion led the choice of the sentences reported in the text?

Statistical analyses are not usually used  when reporting qualitative finding. In line 238 of page 6 it is clearly mentioned that a thematic content analysis approach was used to analyse the responses [28].  The quotes cited were drawn from different parents to reflect each theme.   

Regarding the quantitative analysis, I suggest to summarized several results in a table. It will provide the results more readable.

We are sorry that we could not find a proper way of reporting the findings using a table, because of the possible repetition on the finding in the text while comparisons and percentages at different stages needed to be reported.

Furthermore, I believe that the way to report the chi-squared and t-test results were not correct. For example, Chi sq (df) = value; p = value. Also, when author reported the t-test results, they removed the df value.

The reporting style changes  

D. Discussion and conclusion section

In the conclusion section was missed the prevalence rate of the autism and references.

A prevalence rate of ASD in Iran has been added to the country profile and reference:

The prevalence of ASD in Iranian children was reported to be 6.26 per 10,000 which is lower than that reported for some Western nations, but in line with rates from other countries [30].

-Samadi SA, Mahmoodizadeh A, McConkey R. A national study of the prevalence of autism among five-year-old children in Iran. Autism. 2012 Jan;16(1):5-14.

In general, I suggest to revise the punctuation in all text. For example, there are several typos in the use of bracket, point, and semicolon.

Manuscript was revised for correcting typos and some corrections made

Reviewer 2 Report

The paper enhances then the activity of the Iranian Social Welfare Organization (ISWO), which provides at least 110 daycare centers for children with ASD in Iran across the 31 provinces. See lines 88-89. Cause to the global lockdown for the Covid-19 outbreak, Iran has therefore decided to deliver its professional services for healthcare through mobile-based technology. See lines 93-98.

General information on Iranian clinical and daycare services to pre-schoolers with physical and intellectual disabilities is provided by authors at lines 99-116.

The attention of ISWO in developing of both technology and telecommunication services for daycare centers has been actively deployed since years and also strengthened by the recent rapid outbreak of Covid-19. See lines 118-130.

The study then asks principally the feasibility of telepractice for providing services to family caregivers and children with ASD, as well as how to widespread its use among customers and increase its effectiveness. See lines 145-158.

Eight consecutive weeks of telepractice services in 30 daycare centers across the country have constituted the pool of reference of authors for examining features and eventual gaps in the provision of such daycare service to ASD patients. See lines 153-158.

Starting from line 159 until line 192 the authors enlist features of the telepractice has been implemented by the ISWO centers both in forms of groups and individual sessions with both synchronous and asynchronous sessions.

The data are collected from 30 daycare coordinators from ASD centers and 336 caregivers of children with a confirmed 253 diagnosis of ASD through the professionals of ISWO. See lines 251-254.

Descriptive statistics about the population are given in Table 1. at line 258.

Other detailed descriptive statistics are furnished in Table 2. and 3. See lines 269 and 270.

Qualitative findings are exposed starting from line 280. A positive feedback about telepractice is furnished at lines 303-305, with the only disadvantage it caused extra financial costs and imposed also technical problems such as the internet speed. See lines 305-310. Parents who dropped out from telepractice complained about privacy issues such as taking videos of their child and share information online. see lines 318-321.

Quantitative findings starting from line 331 on show Chi sq and df about parental satisfaction with the support course, and demonstrate parents with younger children were more satisfied about telepractice respect to those with older children. See particularly lines 343-347.

Other interesting aspects of telepractice’s outcomes are furnished at lines 352-389, such as (i) age of parents; (ii) caregivers who had assistance with caregiving; (iii) T-test about areas of concern before and after the online support; (iv) caregivers and parental fidelity in implementing the suggested practices; (v) Key persons’ fidelity.

Fidelity is an important factor for methodological rigor and high fidelity has been linked with optimal outcomes (Penn et al. 2007; Symes et al. 2006; Whiteford et al. 2012. In Ferguson, J., Craig, E. A., & Dounavi, K. (2019). Telehealth as a Model for Providing Behaviour Analytic Interventions to Individuals with Autism Spectrum Disorder: A Systematic Review. Journal of autism and developmental disorders, 49(2), 582–616. https://doi.org/10.1007/s10803-018-3724-5).

Discussion provided by the authors at lines 390-467 seems to indicate telepractice has concentrated economically the deployment of family home-environments to provide care instead of external buildings at the cost of major expenses.

Conclusions at lines 468-475 show telepractice is a valuable help to ASD general treatment.

REQUEST OF MAJOR CHANGES

  1. Please solve the acronym ASD (Autism Spectre Disorders) to its first appearance also in the abstract at line 17.
  1. Please explain for non-practitioners of your interesting topic what is a “Key person” such as those mentioned at lines 383-389.
  1. I would expect from your own research to be shown the use of some techniques—like social narratives, technology-aided instruction and video modeling—through telepractice. Interventions including peer-mediated instruction, parent-implemented intervention and pivotal response training require many indirect approaches, so that to know better which kind of telehealth the ISWO has put into practice could help to arise better conclusions at the end of your paper.
  1. It is known people with ASD struggle with a variety of behaviors like joint attention, verbal and nonverbal communication, restricted interests and routines, and high sensitivity to sensory input. What about to inquire into families’ need to live in a more agreeable neighborhood with sensitive condominiums a/o neighbors as well as the enhancing in the neighborhood of alternative measures of treatment for autism alike pet and occupational therapies?
  1. You mention “coronavirus” in your keywords, but there is so few about the disease in your paper! It is merely mentioned for underlining the need to close daycare centers by the side of the Iranian Healthy Administration. What about to inquire whether coronavirus has modified dietary approaches to autism (being this one another alternative treatment technique for easing autism disorders)?

Has that eased or in the contrary put into danger the application of daycare treatment to ASD patients?

  1. Has the technology literacy of caregivers been enhanced by Covid-19 outbreak or in the contrary lessened because of the lacking opportunities to exchange with others lively on tech issues and also buy them in the main city shops?

On another side has coronavirus put into challenge family relationships between fathers and mothers that can put into danger the health of ASD patients?

  1. Please check at least one of these strands of research and set up better conclusions.

With Kind Regards,  

Author Response

Reviewer 2

We thank you the reviewer for presenting the study summary in a precise and applicable way.

REQUEST OF MAJOR CHANGES

Reviewer 2

Please solve the acronym ASD (Autism Spectre Disorders) to its first appearance also in the abstract at line 17.

Autism Spectrum Disorders is stated in the abstract. Also in the text page 2 line 51.

Please explain for non-practitioners of your interesting topic what is a “Key person” such as those mentioned at lines 383-389.

In the bottom of page 4 line 175 to 180 we have explained the role of the key person:

Each participating center nominated a key person as the main coordinator of the center’s telepractice.  In the most instances, this was a person with the required qualification to supervise the daycare center daily services.  During the lockdown, other center staff were involved with caregivers and children on a scheduled daily basis for routine contact but the key person's responsibility was the coordination, supervision and monitoring the prepared online telepractice program 

I would expect from your own research to be shown the use of some techniques—like social narratives, technology-aided instruction and video modeling—through telepractice. Interventions including peer-mediated instruction, parent-implemented intervention and pivotal response training require many indirect approaches, so that to know better which kind of telehealth the ISWO has put into practice could help to arise better conclusions at the end of your paper.

Following paragraph was added to page 5 line 206:

Parents were guided in the use  of structured teaching, behavioral approaches and environment modifications which were adapted to the child’s communication and sensory  preferences  based on a parent-implemented intervention perspective. In addition features of the  intervention are further described in lines 217 to 229. 

It is known people with ASD struggle with a variety of behaviors like joint attention, verbal and nonverbal communication, restricted interests and routines, and high sensitivity to sensory input. What about to inquire into families’ need to live in a more agreeable neighborhood with sensitive condominiums a/o neighbors as well as the enhancing in the neighborhood of alternative measures of treatment for autism alike pet and occupational therapies?

The short course was provided to assist caregivers during an eight-week COVID19 lockdown of the daycare center and did not mean to be a substitution for other advice and guidance that the   established services had or could provide to parents. Of course the range of the services to cover different aspects of the needs of this group of children and caregivers as you mentioned is a wide range.

You mention “coronavirus” in your keywords, but there is so few about the disease in your paper! It is merely mentioned for underlining the need to close daycare centers by the side of the Iranian Healthy Administration. What about to inquire whether coronavirus has modified dietary approaches to autism (being this one another alternative treatment technique for easing autism disorders)?

Has that eased or in the contrary put into danger the application of daycare treatment to ASD patients?

Has the technology literacy of caregivers been enhanced by Covid-19 outbreak or in the contrary lessened because of the lacking opportunities to exchange with others lively on tech issues and also buy them in the main city shops?

On another side has coronavirus put into challenge family relationships between fathers and mothers that can put into danger the health of ASD patients?

Please check at least one of these strands of research and set up better conclusions.

We appreciate the comment but there is a need to stress  that this was a focused, feasibility study on providing a parent-implemented, home-based intervention for children with ASD.  It was not a study  of the effects of Coronavirus or technological literacy  important as these might be.  Nor could we address family issues and the impact on the ‘health of ASD patients’.  We agree with the reviewer that further research is needed on these issues and not necessarily in the context of telepractice. 

However the conclusions to the study have been updated and the following conclusion was added:

Further research is needed on the issue of COVID-19 and its impacts on children with ASD, their caregivers, support and service or possible alternative treatments not necessarily in the context of telepractice.  

Round 2

Reviewer 1 Report

Dear authors, 

I have read your revised paper and I approve all the improvements you have made.

Reviewer 2 Report

Dear Authors,

I have read you have revised here and there your nice paper, and then it seems to me rather improved!!!

Kind Regards.